# Sustainable Management of Olive Orchard Nutrition: A Review

**Isaac Zipori, Ran Erel, Uri Yermiyahu, Alon Ben-Gal and Arnon Dag ***

Gilat Research Center, Agricultural Research Organization, Gilat 85280, Israel; matabsor@agri.gov.il (I.Z.); ranerel@agri.gov.il (R.E.); uri4@agri.gov.il (U.Y.); bengal@agri.gov.il (A.B.-G.)

**\*** Correspondence: arnondag@agri.gov.il

**Abstract:** Intensification of olive orchard management entails increased use of fertilizers, especially nitrogen, phosphorus, and potassium. In this review, plant responses to nutritional aspects, as well as environmental considerations, are discussed. Nutrient deficiency impairs production, whereas over-fertilization may reduce yields and oil quality, and increase environmental hazards and production costs. The effect of irrigation on nutrient availability and uptake is very significant. Application of organic matter (e.g., manure, compost) and cover crops can serve as substitutes for mineral fertilization with additional benefits to soil properties. Recycling of the pruned orchard material, olive pomace and olive mill wastewater, as well as the use of recycled wastewater for irrigation, are all potentially beneficial to olive orchard sustainability, but present the risk of environmental pollution. Some considerations regarding optimization of olive orchard nutrition are discussed.

**Keywords:** *Olea europaea* L.; fertilization; environmental pollution; sustainability; oil quality; nitrogen; phosphorus; potassium

## 1. Introduction

At the close of 2017, olives (*Olea europaea* L.) covered an area of nearly 11 million ha worldwide [1], with more than 90% of that area concentrated in the Mediterranean Basin, characterized by cold, wet winters and hot, dry summers [2]. In the last few decades, olive cultivation has shifted from traditional, extensive, widely spaced, and rain-fed orchards to intensive, closely spaced, and irrigated ones, leading to an increase in oil production [3]. This increase in oil production has been accompanied by only a minor increase in cultivated area, and can mostly be attributed to management intensification and a rise in yield per unit growing area [4]. Plant nutrition is an essential part of orchard management, especially under intensive cultivation. The olive tree, like most other plants, is composed of 20 elements: C, H, O, N, P, K, Ca, Mg, S, Fe, Zn, Mn, Mo, Cu, B, Ni, Si, Co, Na, and Cl, all essential for proper plant development and production [5]. While $CO_2$ is assimilated from the air, providing plants with their C and some O needs, all other nutrients are absorbed from the soil solution. Large amounts of these nutrients are removed from olive orchards with the fruit [6,7] and, in most agro-systems, with pruned material [8]. Consequently, minerals are depleted from the soil and must be replenished if the trees are to be maintained at an adequate nutritional status. Of the 17 nutrients absorbed from the soil solution, N, P, and K are taken up in the largest amounts and are the most important in terms of both orchard nutrition and fertilization [3] and their impact on the environment, especially under intensive cultivation [9]. Ca, Mg, and S appear in olive plant tissues in relatively large amounts, but rarely need to be applied, as the amounts supplied from the soil and water are usually sufficient. Na and Cl are required by olives in very small amounts, and reference to these nutrients is usually made from the point of view of their negative effect on plant

performance, as they are the major minerals involved in salinity stress [10]. Fe, Zn, Mn, Mo, and Cu are microelements, required in very small amounts [11]. Deficiencies or over-fertilization of these microelements in olives are uncommon [12,13]. Deficiency of one of the essential nutrients will adversely affect plant performance. Conversely, over-fertilization with nutrients may have a negative impact on plant performance as well as on the environment. The objective of the current review, was to discuss the three most important nutrients—N, P, and K—and their effects on the sustainable management of intensive olive orchards.

## 2. Nitrogen, Phosphorus, and Potassium in Olive Orchards

Intensification of olive orchards might very easily lead to over-fertilization and to environmental pollution. There are large differences in potential nutrient removal from olive orchards between extensive, non-irrigated vs. intensive, irrigated systems. Angelo-Rodriguez et al. [11] measured nutrient removal from a traditional, non-irrigated orchard by both fruit and pruned material; annual amounts of removed N, P, and K were 9.7, 1.8, and 9.8 kg ha$^{-1}$, respectively, while local fertilization recommendations were more than 20-fold higher, up to 130 and 240 kg ha$^{-1}$ of N and K, respectively. Erel et al. [3] estimated fertilization rates of N and K under intensive cultivation in Israel to be about 10-fold higher than the amounts removed by the fruit. If the pruned material forms roughly 50% of the total biomass (fruit and vegetative parts) removed from the orchard [8], these fertilization rates are still fivefold higher than the total amounts removed from orchards by both fruit and vegetative material. In an experimental platform described by Haberman et al. [14], control trees received annual rates of 150, 30, and 250 kg ha$^{-1}$ N, P, and K, respectively. Both fruit yield and pruned material were weighed and analyzed for N, P and K concentrations and the results are presented in Table 1. These results are in agreement with estimated calculations made by Erel et al. [3], based on data presented by Dag et al. [15]. The trees in that experiment were first pruned in 2014 and olive paste analysis was not done in 2016. This is why for pruned material there are only three years of data available and for olive paste—five years. However, the average values provide a quantitative estimation of the removed nutrients.

**Table 1.** N, P, and K concentrations and amounts removed from an intensive, commercially fertilized olive orchard.

| | | Concentration in DM (%) | | | Amounts Removed (kg ha$^{-1}$) | | |
|---|---|---|---|---|---|---|---|
| **Pruned Material** | | | | | | | |
| **Year** | **Weight (kg ha$^{-1}$)** | **N** | **P** | **K** | **N** | **P** | **K** |
| 2014 | 10288 | 0.51 | 0.053 | 0.47 | 53.4 | 5.7 | 49.8 |
| 2015 | 7974 | 0.62 | 0.060 | 0.51 | 46.3 | 4.6 | 39.2 |
| 2016 | 13564 | 0.63 | 0.079 | 0.71 | 78.3 | 10.3 | 92.6 |
| **Avg.** | **10609** | **0.59** | **0.064** | **0.56** | **59.3** | **6.9** | **60.5** |
| **Fruit** | | | | | | | |
| **Year** | **Weight (kg ha$^{-1}$)** | **N** | **P** | **K** | **N** | **P** | **K** |
| 2011 | 6578 | 0.62 | 0.062 | 1.35 | 41.0 | 4.1 | 88.8 |
| 2012 | 6230 | 0.71 | 0.080 | 1.40 | 44.0 | 5.0 | 87.4 |
| 2013 | 7013 | 0.65 | 0.073 | 1.22 | 45.6 | 5.1 | 85.8 |
| 2014 | 5554 | 0.82 | 0.087 | 1.42 | 45.7 | 4.9 | 78.6 |
| 2015 | 3506 | 0.74 | 0.087 | 1.43 | 25.8 | 3.1 | 50.0 |
| **Avg.** | **5776** | **0.71** | **0.080** | **1.36** | **40.4** | **4.4** | **78.1** |

The average annual amount of N, P, and K removed from the orchard was 99.7, 11.3, and 138.6 kg ha$^{-1}$, respectively. These figures indicate that 70% of the applied N, 40% of applied P and 58% of applied K were recovered in the removed plant parts. This fertilization efficiency is much higher than that described by Angelo-Rodriguez et al. [11], probably due to the high efficiency of the fertigation system used to supply the nutrient needs of the trees. In the case of N, most of the unconsumed applied N will leach below the root zone [9] and part of it will reach the atmosphere as N$_2$O [16]. In

the case of K, most of the unconsumed K will find its way to the adsorbing complex of the soil, unless the cation-exchange capacity of the soil is very low, in which case it will also be leached below the root zone [17,18]. Excessive soil K load may lead to soil dispersion and reduced infiltration rate, as described for sodicity [19,20]. Unconsumed P will be partly fixed and remain in the upper soil layer, where it might be transported by water runoff and, consequently, translocated to undesirable locations such as water bodies [21]. In fertigated (fertilized via the drip-irrigation system) orchards, P can also migrate below the root zone and find its way into groundwater, especially in very sandy or low pH soils. However, since olives are generally irrigated using a deficit-irrigation strategy, this risk is lower than for other crops.

## 3. Rain-Fed and Irrigated Olive Orchards

Olive cultivation occurs in two major and distinct systems: (i) Traditional, extensive, rain-fed orchards and (ii) modern, high-density, intensive, irrigated orchards. Most of the world's olive orchards are rain-fed [2] but their relative contribution to production is low [4]. Plants have to be fertilized in both cultivation systems, but fertilization considerations and risks of potential environmental pollution differ for each. In rain-fed orchards, growth rates and yield levels are much lower than in irrigated systems [22–24] and therefore, fertilizer application rates are generally lower than for irrigated orchards [3,25]. In rain-fed orchards, water is very often the limiting factor for nutrient availability and uptake, since during the long dry summer, availability is reduced [3,13]. Timing of fertilization is unique for each category. In rain-fed orchards, fertilizer is applied to the soil, and the mobilization of nutrients to tree roots is highly dependent on rainfall events. In the Mediterranean climate typical of olive-cultivation regions, organic substances or mineral fertilizers are commonly applied in late winter or early spring, designed to take advantage of rainfall events to transport minerals into the root zone. Insufficient rainfall will leave nutrients out of reach of active roots, and excessive rainfall may lead to significant N losses by leaching or release of gaseous N forms (denitrification). Fertilization during the dry season is usually done by foliar application [25,26]. Ferreira et al. [27] found that repeated foliar applications of N can maintain adequate leaf N levels throughout the growing season. However, foliar spraying generally fails to meet the macro-element requirements of fruit trees [28]. In an experiment carried out by Toscano et al. [29], four annual foliar applications proved to be useful as a complementary activity to soil application, but could not satisfy the full nutritional requirement of the trees over the long run. Nutrient uptake from foliar applications is also affected by tree water status, such that the uptake by water-stressed trees is lower than that by non-stressed ones [30,31]. In irrigated orchards, fertilizers are usually applied simultaneously with irrigation (fertigation) [3], which enables better control of fertilization levels and timing and thus, may enhance fertilizer-use efficiency compared to broadcast application [32,33]. The two different systems also imply differences in potential environmental effects. On the one hand, the amount of nutrients that need to be applied to rain-fed orchards is lower than for irrigated ones. On the other, the grower has less means of controlling the unused residual nutrients, which can contribute to environmental pollution.

## 4. Nitrogen

N is a major essential plant nutrient and the most commonly applied mineral in fertilization programs of horticultural crops [34]. Until a few years ago, there was a debate regarding the significance of annual application of N for olive productivity as part of olive orchard management, including claims that N is generally supplied in excess [35]. This argument was supported mostly by evidence from extensive, non-irrigated orchards [36,37], where olive tree performance responded positively to N fertilization only when trees were grown under harsh N-supply conditions [38,39]. Recent experiments, carried out on intensively cultivated, irrigated olive orchards, revealed a strong positive response of tree performance to N fertilization [14,40–42]. Adequate N fertilization resulted in more vigorous vegetative growth, producing shoots with potential for carrying yields in the subsequent season [14]. In extensive, non-irrigated olive orchards, both soil and foliar N application is common [6]. This practice offers little opportunity to control soil N dynamics. In intensive, irrigated

orchards, fertigation is the common practice, allowing maintenance of high levels of available soil N. In both cultivation systems, over-fertilization with N might lead to yield reduction, by downregulating physiological processes related to flower quality [43] and fruit set [14], or due to some not yet fully understood toxic effect [44,45]. Furthermore, over-fertilization with N adversely affects oil quality, especially via elevated free fatty acid content and reduced polyphenol levels in the oil [15,46]. Reduced polyphenol levels in the fruit affects fruit health by reducing their resistance to fungal infestation, which further impairs oil quality [47]. Figure 1 provides a schematic illustration of the composite response of olive trees to N fertilization.

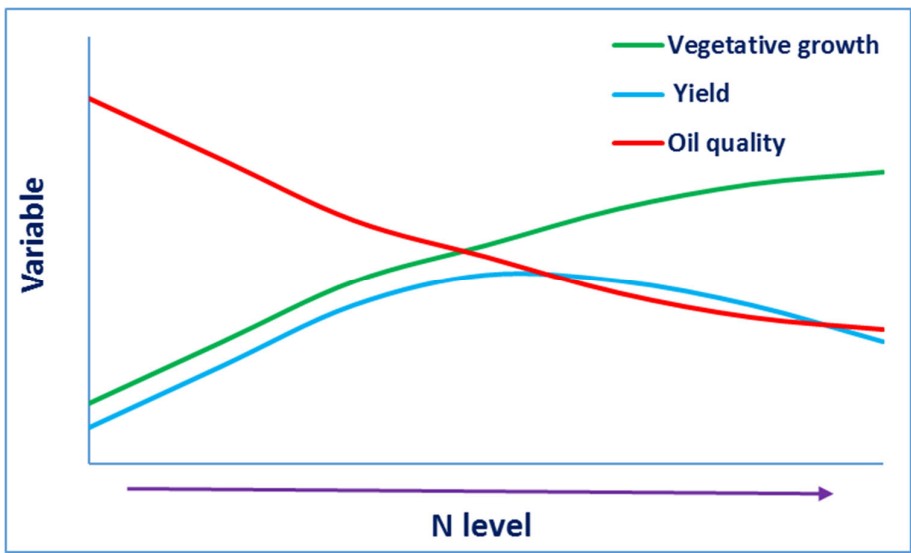

**Figure 1.** Schematic illustration of the response of olive trees to N fertilization.

Another important aspect of over-fertilization with N is environmental pollution. N is applied to olive trees in various ways: Direct soil application as soluble solid compounds, organic matter application, or application of a liquid or soluble form using fertigation or foliar sprays. All of these fertilization practices may lead to elevated soil N levels, mostly in the form of nitrate ($NO_3^-$). $NO_3^-$ is not adsorbed to the soil solid phase, is easily transported below the root zone, and becomes a groundwater pollutant when it is not taken up by plants [9]. Unconsumed N can also be transformed into nitrous oxide ($N_2O$), a greenhouse gas [48]. Optimization has to be reached to balance adequate N supply to the trees with minimal pollution risks. Vilarrasa-Nogué et al. [49] showed that when N was supplied to soil at the relatively low rate of 100 kg ha$^{-1}$ year$^{-1}$, $N_2O$ emission into the atmosphere was negative in most cases. In a 6-year experiment described by Haberman et al. [14], soil samples were taken twice a year: In spring, at the end of the rainy season before the start of fertigation, and in autumn, after the end of fertilizer application before the first rains. Soil $N–NO_3$ concentrations in saturated paste extracts are presented in Figure 2, indicating that at the low N-application rates, almost all of the applied N was consumed and the amount leached was minimal. However, under high application rates, large amounts of N leached below the root zone.

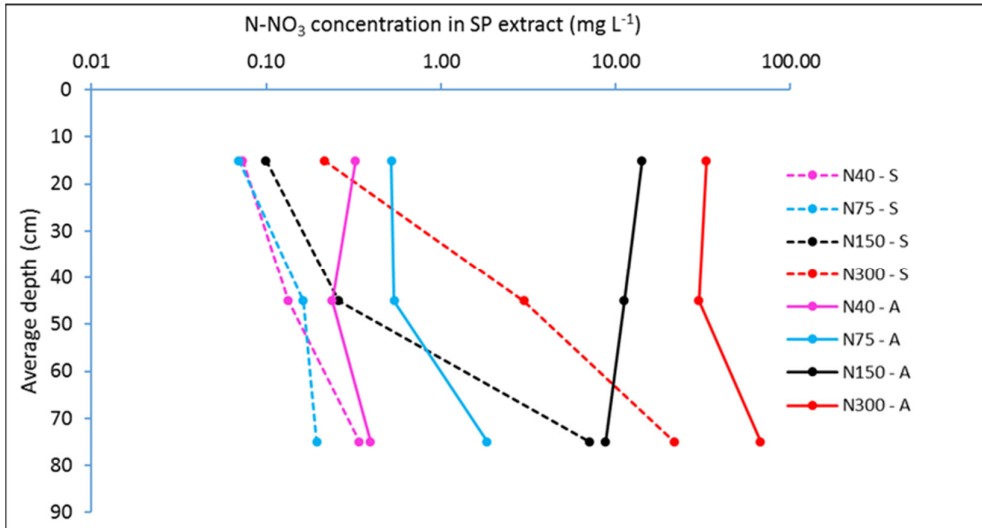

**Figure 2.** N–NO₃ concentrations in soil paste (SP) extracts from the 0–90 cm soil profile under four annual N-application rates: 40, 75, 150, and 300 kg ha⁻¹, in spring (S) and in autumn (A). The results are averages of 5 years.

The use of nitrification inhibitors has been suggested to reduce both leaching and release of N to the atmosphere. However, this approach is debatable as the nitrification inhibitors might increase ammonia volatilization and, subsequently, $N_2O$ emission [16].

## 5. Phosphorus

Until recently, knowledge regarding the role of P in olive cultivation was very limited [43,50], and common practice suggested no P fertilization as long as leaf P levels were above 0.1%. Publications concerning the direct effect of P on olive productivity have been published only in the last decade [41,42,51–53]. The lack of information is likely a result of the common belief that olive trees take up P very efficiently [25,26,50], due to their extensive root systems and symbiosis with mycorrhiza fungi [54,55]. Visual symptoms of P deficiency are rare and thus P is not commonly applied as a fertilizer [25,26,50,56]. Soil P mobility is very low, and most P uptake occurs by root interception [57]. Therefore, in contrast to broadcast application, P fertigation improves P availability and enhances the plants' potential to take up P rapidly when required [13,33]. Reports from controlled container experiments found that increased P levels have a direct positive effect on the whole reproductive cycle and contribute to an increase in yield [41,51].

Due to its low soil mobility, P rarely reaches the groundwater as a pollutant. Most P pollution is caused by mass transport at the soil surface as a result of soil erosion and water runoff from agricultural, fertilized fields [58]. In this way, P can reach water bodies such as lakes and rivers, adversely affecting their biological equilibrium and enhancing eutrophication [59]. However, P fertigation uses soluble P fertilizers and increases P mobility. In the long-term experiment carried out in Israel and described by Haberman et al. [14], the experimental platform described for N (Section 4) was used to study the response of olive trees to P nutrition. Three P levels: 0, 80, and 160 kg $P_2O_5$ ha⁻¹, were applied annually and soil was sampled in the 6th year. The results, presented in Figure 3, show P accumulation in the 0–60 cm layer at the end of the irrigation season (autumn) as a result of deficit irrigation, and much lower levels in spring, at the end of the rainy season, during which no fertilizer was applied (Yermiyahu U., unpublished data). This strongly indicates that P leached below the upper soil layer, increasing the risk of P contamination of the groundwater. Although P is considered highly immobile in soils [60], several studies have shown that P availability and mobility increase with increasing soil water content, especially when fertigation is applied [13,60].

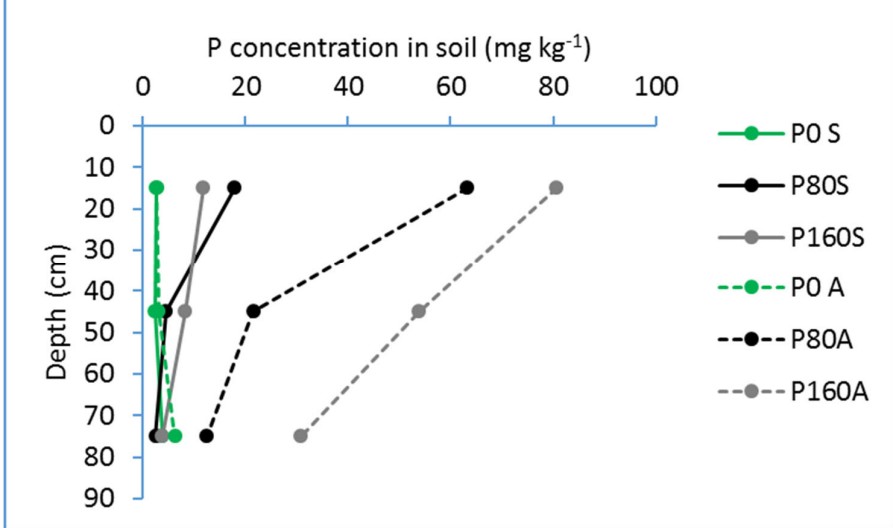

**Figure 3.** Soil P (Olsen) concentrations in autumn (A) and spring (S) in a fertilization field experiment in olives. P0, P80, and P160 represent 0, 80, and 160 kg $P_2O_5$ $ha^{-1}$ $year^{-1}$, respectively.

## 6. Potassium

Potassium is considered to be one of the most important minerals in olive nutrition [50,61], probably due to its high concentrations in the fruit flesh [50,62], and because K deficiencies are relatively common in rain-fed olive orchards [56]. In the past, some reports relating K in rain-fed olive trees to yield were published and were subsequently used as the basis for fertilization recommendations [26,61]. There are additional reports regarding K deficiencies in olives around the Mediterranean basin [63], but they also refer to rain-fed and not to irrigated orchards. It seems that irrigation in itself increases K availability, without any supplemental K fertilization [64]. Zipori et al. [13] also found an increase in K uptake with increasing irrigation level. K mobility and availability for plant uptake is higher under fertigation compared to broadcast application [33]. However, this elevated uptake cannot be attributed solely to the increase in soil K availability resulting from higher K translocation from the adsorbing complex into the soil solution, as uptake of K from foliar application also depends on water availability [31].

Although K is considered to have an important role in plant water balance and carbohydrate assimilation in olives [65], Erel et al. [66] found no correlation between K levels and drought tolerance, including stomatal control mechanisms. However, K deficiency was found to impair photoprotection mechanisms due to a reduction in photosynthetic and photorespiratory capacity [67].

Information on olive response to K fertilization seems to be contradictory. On the one hand, K concentrations in the fruit and pruned material are high, and consequently, annual K offtake is higher than for any other nutrient (Table 1). On the other hand, many studies found no response of vegetative growth or yield to K fertilization, even though leaf K concentrations increased with increasing K levels [30,41,50,68]. In the 6-year field experiment described by Haberman et al. [14], there was no difference in vegetative development between trees not fertilized with K for 6 years and trees fertilized with 300 kg $K_2O$ $ha^{-1}$ annually, which is in agreement with the above mentioned reports. However, yield was significantly higher in the fertilized trees, a result of more intense flowering and higher fruit number per tree (Yermiyahu, U., unpublished data). This response to K fertilization occurred even though leaf K concentration in the non-fertilized trees was never below the deficiency threshold of 0.4% and reached 0.82% after 6 years, which is close to the 0.8% value considered sufficient [50]. This result strongly indicates that sufficiency threshold values for intensive, irrigated orchards should be revised and updated, and that adoption of threshold values from rain-fed orchards for intensive, irrigated ones needs to be reconsidered in the case of K. There is an increase in K availability and uptake as a response to irrigation only, but apparently this increase cannot satisfy plant needs; when trees are well supplied with water, K nutrition may become a

limiting factor, especially due to the large biomass removed with the fruit and pruning material in intensive orchards. In this case, the available soil-K pools (adsorbed K) are depleted from year to year until eventually, K starvation limits fruit yield. Erel et al. [66] found that Na could partially substitute K when supply of the latter was limited, but the effect of using Na as a substitute for K on yields was not tested.

K is the least important macro-element with respect to soil and water pollution [69]. Over-fertilization with K can result in elevated percentages of exchangeable K [18]. Savci [70] stated that over-fertilization with K can impair soil structure. However, Levy et al. [17] found that aggregate stability, a measure of soil structure, increased with increasing levels of K, and that soil hydraulic conductivity was not impaired by elevated soil K concentrations when organic compounds were added to the soil. Over-fertilization with K can affect salinity indirectly as, in general, the source for K is KCl.

## 7. Organic Material

The source for nutrients can be either mineral fertilizers or organic materials. Relevant organic materials incorporated into olive orchards include compost, raw organic manure, olive mill wastewater (OMW) and chopped pruned material. Direct application of non-composted animal manure, while common, is not recommended due to the risk of spreading pathogens, parasites and weed seeds, all of which are eliminated during the composting process [71]. For this reason, compost is the preferred substance for organic nutrition. Different composts have varying concentrations of nutrients, depending on the source materials used for their preparation [72]. Concentrations of 1.4–2.5% total N, around 0.3–1.0% P and 2.1–2.9% K in the dry matter are common [72,73]. Most composts are prepared from varying amounts of animal manure, a bulking agent such as wheat straw [74,75] when needed (e.g., in the case of slurries), and very often, materials from other sources, such as olive pomace from two- or three-phase olive mills, OMW, grape husk, etc. [76]. In an experiment carried out by Cayuela et al. [72], 40 kg compost tree$^{-1}$ (14.3 tonnes ha$^{-1}$) was applied in a drip-irrigated orchard. The potential contribution of the compost to N, P and K tree nutrition in that experiment was 75–84, 11–17, and 100–120 kg ha$^{-1}$, respectively. These amounts are roughly 50% of the recommended levels of 150, 30, and 250 kg ha$^{-1}$ of N, P and K, respectively [3]. Apart from its effect as a source for nutrients, compost has a positive effect on many other soil properties. In a bio-organic orchard fertilized with compost, organic matter content increased from 1.8% to 5% and cation-exchange capacity from 8 to 17 meq 100 g$^{-1}$; soil hydraulic conductivity increased as well, resulting in higher soil water retention [72]. Compost application increases the humic fraction of the soil and improves aggregate stability [77]. Compost application also increases soil organic C content [78]. These latter authors found that an increase of 50% in soil organic C results in doubling the macroporosity, from 4% to 9%, and increasing hydraulic conductivity from 0.5 to almost 7 mm h$^{-1}$ and soil water content by 25% in the 0–200 cm layer.

An important environmental aspect of compost application is the recycling of animal manure. Animal husbandry produces large amounts of manure that might become an environmental pollutant if not treated properly. The incorporation of this material into composts is beneficial for both parties: Animal husbandry and horticulture [76].

A potential disadvantage of compost application is the risk of N migration out of the field with runoff or below the root zone when application is followed by heavy rain or irrigation [79]. However, most of the N in compost is in organic form, which is less liable to migrate downward through the soil with water. Mineral N content in compost is usually one order of magnitude lower than total N [73]. Therefore, this potential disadvantage is expected to become a problem only if extremely high amounts of compost are applied locally.

As olive orchards are often planted on marginal soils, characterized by low water holding capacity and poor nutrient levels, tree development and performance are sometimes impaired, leading to low yields. In many orchards, growers apply compost in the planting hole to improve tree performance in the first years [80]. However, the effect of this practice fades after a few years, because of both decay of the compost and root development outside the planting hole. In Israel, growers use

a different technique: They prepare a shallow trench, 30–40 cm wide and 30 cm deep, parallel to the tree row, at a distance of 80–100 cm from the tree line. Compost is applied in the trench and then covered with a thin soil layer (Figure 4). The drip lines are then placed on the soil and compost-covered trench. For subsurface drip irrigation, the drip lines are placed either at the bottom of the compost trench or above the compost and below the soil layer covering it. The result is proliferation of roots inside the trench volume and improvement of tree performance in the orchard. The compost trench is usually renewed every 2–3 years. This method is used not only for olives but also with other fruit tree crops in arboriculture, and in intensive vegetable cultivation [81,82]. The technique can potentially address problems arising from sodic, non-aerated soils with extremely high sodium adsorption ratio (SAR) values or high pH values, and infertile calcareous soils, and contribute to satisfying nutritional demands in intensive olive orchards [83]. It is quite obvious that the compost trenches contribute to both chemical–nutritional aspects as well as chemical–physical aspects, but the relative contribution to each group is difficult to assess.

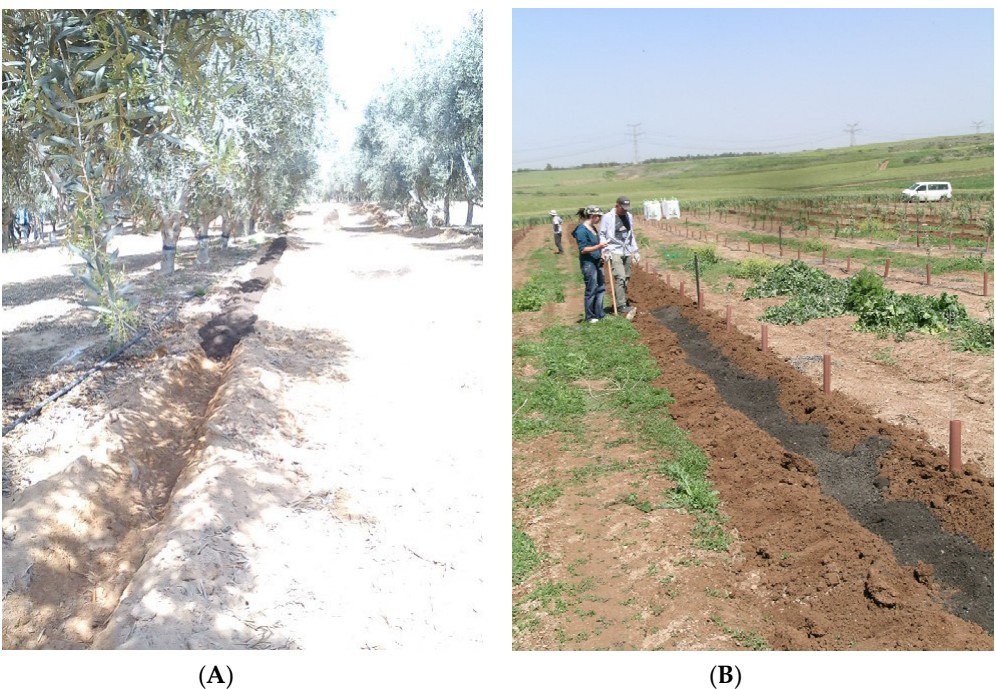

(**A**)                                                                                    (**B**)

**Figure 4.** Compost trench installation in mature (**A**) and young (**B**) olive orchards.

## 8. Cover Crops

The use of cover crops (Figure 5) in olive orchards is usually related to their ability to reduce the risks of soil erosion and runoff, especially when cultivating on sloped terrains [58,84–88]. However, there is no consensus regarding the nutritional aspects of cover crops in general, despite their contribution to soil fertility [89,90], as they compete with the olive trees, at least to a certain extent, for nutrients and water [91,92]. Cover crops can be either perennial or annual and can be natural vegetation or purposefully sown [93]. Annual cover crops are usually maintained on the surface during the rainy period and subsequently eliminated from the soil surface by either herbicides or incorporating them into the soil [89]. The use of cover crops reduces nutrient losses with runoff and through soil mass transport [90], and thus contributes to both orchard soil fertility through better nutrient retention and reduction of environmental pollution [93]. The direct effect of cover crops on soil fertility and olive tree nutritional status depends on many factors. The common concept is that leguminous cover crops will contribute more N to the soil than non-N-fixing crops [92,94]. However, Rodrigues et al. [95] found only a slight contribution of leguminous residues to soil mineral N, and little N contribution to the trees, stating that most of the N was lost, probably by denitrification. The

C:N ratio of the cover crop residues strongly affects their contribution to the N balance. When this ratio is high, N immobilization is the dominant process whereas when it is low, N mineralization occurs [96]. Trinsoutrot et al. [97] found that only residues with a C:N ratio lower than 24 induce a surplus of mineral N, and suggested the use of this ratio as an indicator for the potential contribution of plant residues to mineral soil N. Decomposition rates of cover crops are higher when the crop is incorporated into the soil compared to when they are left on the surface [93].

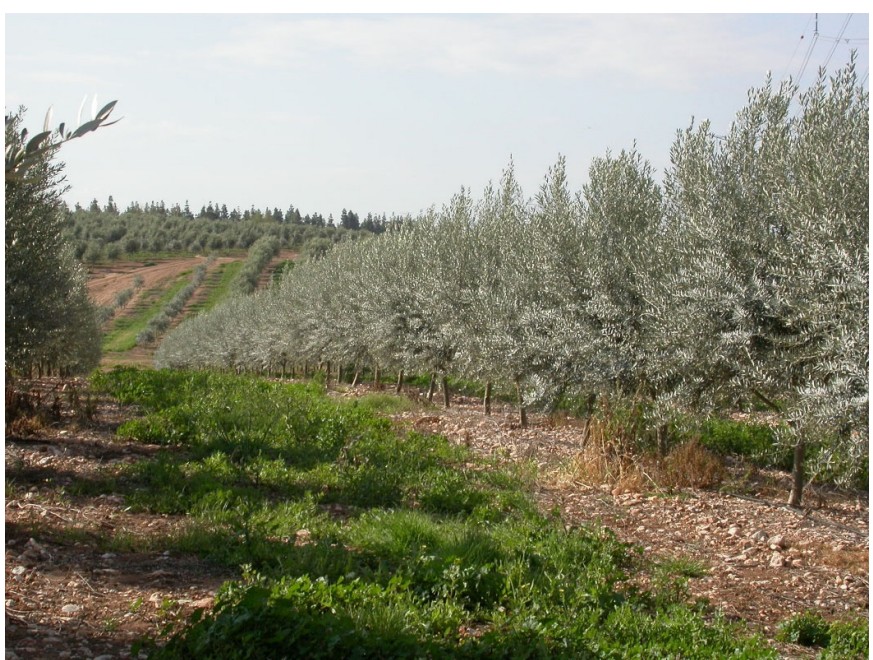

**Figure 5.** Cover crop (natural vegetation) in a commercial olive orchard in a hilly area in Israel.

Most studies regarding cover crops and olive orchards show a positive effect on the environment through reduction of nutrient losses, as well as improved soil fertility. Enhancement of soil fertility by cover crops is due to improved soil physical properties, increased water retention and enriched nutritional properties [85]. When a cover crop policy is adopted, it is usually accompanied by no-tillage practices, which have their own benefits with respect to tree performance, soil fertility and conservation [86,89,90], as well as on $CO_2$ emission to the atmosphere. The practice of cover crops accompanied by no-tillage in intensive olive orchards has more advantages than disadvantages, and can be integrated as a key component in sustainable olive orchard management [91,98].

## 9. Recycling and Fertilization

Some nutrient recycling occurs naturally. Dropped leaves and fruit form a soil layer with higher organic matter content below the trees (Figure 6). The life span of an olive leaf is approximately 2 years [99]. Toward the end of this period, leaf-aging processes begin, including nutrient translocation to younger leaves [100] and subsequently, leaf drop. The dropped leaves are a potential source of nutrients as they eventually decompose. However, it is extremely difficult to assess the quantitative nutritional contribution of these leaves to the orchard as decomposition rates depend on environmental factors and management practices.

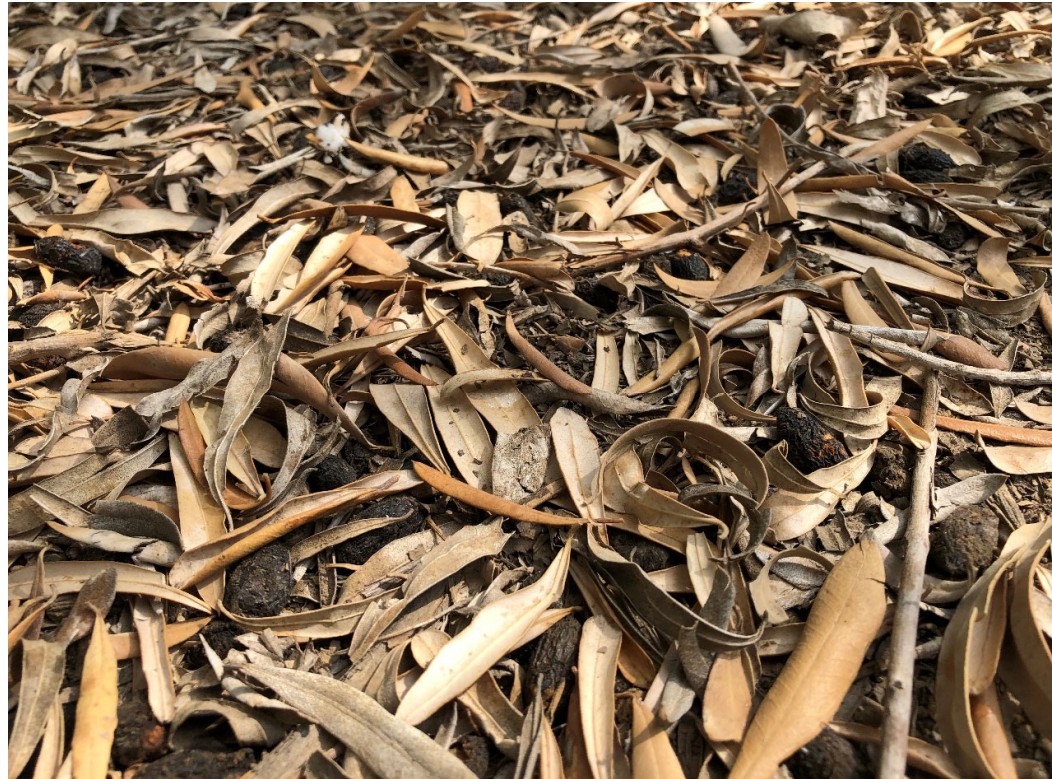

**Figure 6.** A layer of leaves and fruit under the canopy in a mature (15-year-old) olive orchard at the Gilat Research Center in Israel.

Gómez-Muñoz et al. [93] found that about 80% of P and K in plant residues are released after 1 year and that decomposition rates on the soil surface are lower than for incorporated material. At the end of an experiment described by Erel et al. [41], total leaf dry weight on the trees was 32 kg tree$^{-1}$, which is equivalent to 11.2 ton ha$^{-1}$ in an intensive orchard with a planting density of 350 trees ha$^{-1}$. If concentrations of 1.6%, 0.1%, and 1.0% are taken into account for N, P and K, respectively, the total amount of nutrients in the leaves reaches 180, 11 and 112 kg ha$^{-1}$ of N, P and K, respectively. If 50% annual leaf replacement is considered, the potential annual contribution of leaves to the orchard is 90, 5.5, and 56 kg ha$^{-1}$ of N, P and K, respectively. Some of this potential contribution occurs prior to leaf detachment. During leaf senescence, nutrients are translocated from the old (previous season) leaves to the young (current season), developing ones. In the experimental platform described by Haberman et al. [14] (Section 4), nutrient concentrations in the concurrent year's diagnostic leaves, sampled in July, were compared to their concentration in 1-year-old leaves, sampled at the same time, over 3 successive years, from the control treatment, receiving 150, 80, and 300 kg ha$^{-1}$ N, $P_2O_5$ and $K_2O$ annually, respectively. N, P and K concentrations in the old leaves were, on average, 25%, 12%, and 20% lower than in the young leaves, respectively (Table 2).

**Table 2.** N, P, and K concentrations in dry matter of young (concurrent year's) and old (previous year's) leaves. Significant differences between young and old leaves for each nutrient are indicated by different letters (Tukey-Kramer HSD, $p < 0.05$).

| Year | Avg. Yield (kg tree$^{-1}$) | N (%) Young | N (%) Old | P (%) Young | P (%) Old | K (%) Young | K (%) Old |
|---|---|---|---|---|---|---|---|
| 2014 | 37.6 | 1.70a | 1.13b | 0.125A | 0.086B | 1.12*A* | 0.69*B* |
| 2015 | 25.0 | 1.57a | 1.46a | 0.132A | 0.123A | 1.17*A* | 1.19*A* |
| 2016 | 31.4 | 1.42a | 1.07b | 0.105A | 0.093A | 0.95*A* | 0.77*B* |

It can be assumed that these rates continue to increase with leaf senescence until leaf detachment. It is worth noting that differences in N, P, and K concentrations between young and old leaves were linked to fruit yield. In years with relatively high yields (2014 and 2016), the differences were significant, whereas in a low-yield year (2015), they were not. This calculation can serve as a first approximation of the naturally occurring recycling processes in olive orchards. As already noted in Table 1, large amounts of nutrients are removed from the orchard with the fruit and the pruned material. The N, P, and K removed with the fruit are found in the olive pomace and the OMW, as discussed in Section 10. The pruned material can be removed from the orchard, but then it raises challenges for waste disposal and sometimes, issues of phytosanitation, unless chopped and incorporated into compost. Another common, and possibly superior solution is chopping this material on site [101]. In the long run, chopped pruned material decomposes, increases the organic matter content of the soil, and can potentially contribute to orchard nutrition via mineralization processes. The data presented in Table 1 indicate that the potential annual contribution of chopped pruned material can reach 60, 7, and 60 kg ha$^{-1}$ N, P, and K, respectively. Another advantage of chopped pruned material is its mulching effect, reducing water losses and soil erosion, increasing soil organic matter content, improving soil structure, and reducing weed growth [101]. Figure 7 shows an example of a mature intensive olive orchard after chopping of the pruned material.

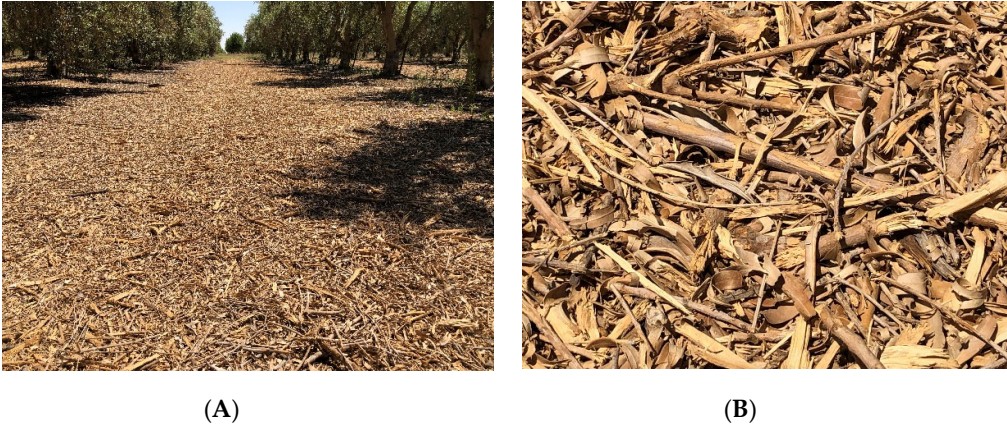

(**A**)  (**B**)

**Figure 7.** A mature olive orchard after chopping of pruned material (**A**) and a close-up view of the chopped material (**B**), Gilat Reseach Center, Israel.

## 10. Recycling of Olive Pomace and OMW

The olive oil extraction industry produces large amounts of olive pomace and OMW, which are potential environmental pollutants. Olive pomace is easier to handle than OMW, with various options for recycling, the most common being its incorporation in compost, together with organic manure and some bulking agents when required [102]. Pomace can also be dried and burned with a high energetic value or serve as livestock feed after appropriate treatment [103]. OMW is more difficult to handle. This byproduct, produced in large amounts over a relatively short time, cannot be

introduced into sewage treatment plants shared with municipal or industrial players due its high biological oxygen demand (BOD), fat and polyphenol concentrations. In some cases, OMW is stored in large pools and left to dry in the sun until the solid leftovers can be incorporated into composts, but this approach requires large storage and evaporation ponds and tends to be an environmental nuisance due to unpleasant odors. The most common solution today is to spread OMW on the soil surface in olive orchards (Figure 8), in limited amounts of 50–80 m$^3$ ha$^{-1}$ year$^{-1}$ [18]. Potential risks of in-situ orchard application of OMW include negative effects on soil characteristics [104] or soil fauna due to phytotoxicity [105] and transport out of the orchard and into natural water sources during erosion events. Nevertheless, recent studies [17,18] have shown that controlled application of OMW to olive orchards over several years has no long-term adverse effects on soil properties or tree performance. Saadi et al. [106] found quick recovery of soil microbial activity after OMW application and recommended this procedure as safe. In addition, OMW can provide significant amounts of K and P to the soil and partially replace fertilization with these nutrients.

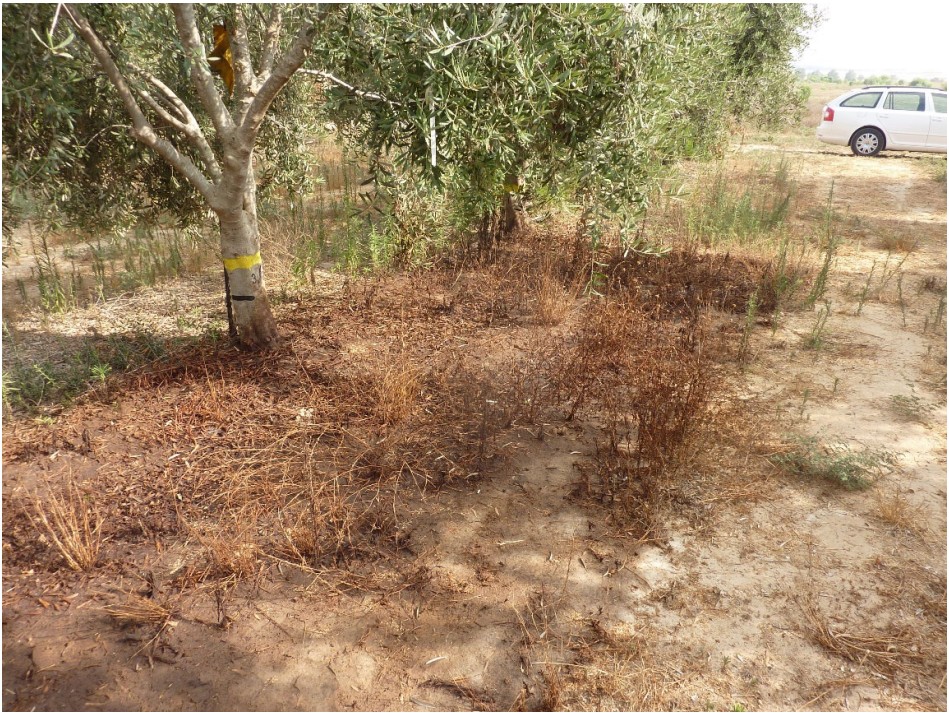

**Figure 8.** An olive plot, 6 months after olive mill wastewater (OMW) application (50 m$^3$ ha$^{-1}$). Gilat Research Center, Israel.

Zipori et al. [18] found that at an annual OMW application of 50 m$^3$ ha$^{-1}$, the whole K requirement and 30% of the P requirement of the orchard were satisfied, suggesting substantial savings on chemical fertilizers. Controlled application of OMW in olive orchards seems to be a sustainable practice, especially if it is followed by shallow tillage to reduce soil hydrophobicity and improve water infiltration [17]. However, there is always the risk of a negative impact of OMW on soil properties. If an intensive olive orchard produces an average yield of 13 t ha$^{-1}$ year$^{-1}$ and the amount of OMW generated during the oil-extraction process is 1.5 m$^3$ t$^{-1}$ [106], then the total annual amount of OMW generated is 20 m$^3$ ha$^{-1}$. In most countries, the permissible annual amount of applied OMW is 50–80 m$^3$ ha$^{-1}$ [107]. This means that a given plot will be exposed to OMW once every 3–4 years, thus further reducing possible negative effects but being less effective in terms of nutrient supply and fertilizer replacement.

## 11. Irrigation with Recycled Wastewater as a Source of Nutrients

Global demand for water for domestic, industrial and agriculture uses is continuously rising, as a result of population growth and standard of living. Competition for high-quality water resources is prominent in water-scarce regions, where irrigation is essential to agricultural expansion and success. The need to treat and dispose of increasing quantities of sewage on the one hand, and the rising demand for irrigation water on the other, stress the importance of effective and sustainable use of recycled wastewater (RWW) [108,109]. In regions where no fresh water is available for olive orchard cultivation, irrigation with RWW has been shown to provide a sustainable alternative and improve yields [110].

While the use of RWW in agriculture can help meet the increasing requirement for water across the agricultural, domestic and industrial sectors [111], irrigation with RWW carries both agronomic and environmental risks that require special consideration [9]. On the positive side, use of RWW allows recycling of both water and nutrients that would otherwise be disposed of in the environment, subsequently contaminating natural water bodies. On the downside, RWW tends to contain high and potentially problematic concentrations of plant growth-inhibiting ions such as Na, B, and Cl [112,113]. High concentration of Na in RWW, relative to those of Ca and Mg, has added potentially hazardous effects due to its contribution to elevated SAR values. Nevertheless, in an 8-year experiment [114], no negative effect on tree performance was detected as a result of irrigation with RWW in comparison to irrigation with fresh water. Moreover, even when no fertilization was applied for 5 years, irrigation with RWW was sufficient to satisfy the nutritional requirements of the trees, as the annual amounts of N, P, and K supplied by the RWW alone were 124, 34, and 193 kg ha$^{-1}$, respectively. This calculation was based on an analysis of the soluble inorganic constituents of the RWW. However, RWW also contains significant amounts of organic matter. Mineralization rates of this organic matter and its effect on the soil microbiological population are difficult to assess quantitatively. When irrigating with RWW, under low levels of BOD and chemical oxygen demand (COD) (20 and 70 mg L$^{-1}$ O$_2$, respectively), soil microbiological composition and activity were not affected [115]; under high BOD and COD levels (80 and 173 mg L$^{-1}$ O$_2$, respectively), both the composition and activity of the soil's microbiological population were affected [116]. The activity of ammonium-oxidizing bacteria was significantly affected even at the low BOD and COD levels [115]. This elevated microbiological activity transforms organic N into mineral N and thus contributes even more to the potential N supply from RWW. Similar processes probably occur in the transformation of organic P into mineral P. The actual contribution of RWW to olive nutrition is site-specific, depending on the actual concentrations of the individual minerals in the water and the amount of water irrigated. Obviously, when irrigating with RWW, the contribution of nutrients delivered with the water to the trees has to be taken into account and subtracted from the fertilization scheme.

In a study on intensive hedgerow olive cultivation, irrigation with RWW caused salt accumulation during the summer due to the deficit-irrigation policy employed, but the salts were leached each year with the winter rains [9,114]. A negative trend was identified for SAR, which increased slowly and steadily during the 8 years of the experiment in the plots irrigated with RWW, compared to plots irrigated with fresh water. Continuous irrigation with RWW might impair the soil's physical properties [110,114] and measures, such as enrichment with Ca ions by liming, should be considered.

## 12. Effect of Soil pH on Selection of Fertilizers

Most olive orchards are grown on calcareous soils, with a pH higher than 7.0. Under these conditions, microelement availability may be limited [57,117]. One way to overcome this problem is foliar application of microelements, which seems to be more effective than soil application of chelates in these cases [117]. Another approach is to reduce the pH of the root–soil interface with ammonium-based N fertilizers, with or without nitrification inhibitors. When an ammonium ion is absorbed by a plant root, a proton is released to maintain the system's electrical balance and the pH of the root–soil interface is reduced, resulting in a more favorable environment for microelement availability

[118]. In some cases, the expansion of olive growing into other regions of the world has led to olive planting in low-pH soils [24]. To elevate soil pH to values closer to the natural habitat of olives, fertilization with $NO_3$-based fertilizers can be beneficial. However, most low-pH soils are in regions with relatively heavy summer rainfall, which might enhance N migration below the root zone, reduce fertilization efficiency, and increase N pollution risk. This requires the adaptation of a quite sophisticated fertilization regime, e.g., application of fertilizer (in the case of fertigation) toward the end of the irrigation cycle, and application of a small amount in each fertigation cycle so that the risk of pollution is reduced.

## 13. Fertilization Management

Timing of fertilizer application is important. Nutrients have to be available for uptake when the plants need them. Under Mediterranean conditions, tree activity during the cold winter months is reduced to a minimum and therefore, nutrient uptake during these months is practically nonexistent. Moreover, in the winter, the soil is leached by seasonal rains. Nutrient levels can therefore be very low in the springtime [9], when trees begin to bloom and grow and have a high demand for them. As mentioned further in this section, toward harvest, other considerations are involved.

In rain-fed olive orchards, fertilization is based on the application of pre-determined annual doses, and the grower has no control over nutrient availability to the trees. Therefore, most fertilization recommendations are in terms of kilograms per hectare per year. The introduction of fertigation enables the grower to fully control nutrient levels in the irrigation water at any given moment and adjust those levels according to phenological stage and fruit load.

Olives are typically irrigated at deficit levels to optimize yield and oil quality [119,120]. Due to this policy, translocation of nutrients below the root zone during the fertigation season is unlikely to occur under Mediterranean conditions [9]. However, in the last few decades, olive cultivation has expanded to other regions, characterized by summer rains, sometimes in significant amounts [24]. Under such conditions, fertilization regimes must be modified, e.g., frequent broadcasts of solid fertilizers if irrigation is not required at all, or delivery of fertilizers to the soil by applying small-volume irrigation events with high nutrient concentrations. The precise fertilization regime has to be adapted to the prevailing conditions in a given location.

Based on findings from recent studies [15,46], a more regulated fertilization practice has been adopted by many growers in Israel. The total annual amount of N to be applied is based on leaf analysis data, but the whole amount is applied from March–April (the beginning of the irrigation season) to the end of August–beginning of September, when oil accumulation is at its maximum rate. This practice has two purposes: (i) by the time the local late autumn/early winter rains begin, soil N has been substantially depleted by tree consumption and the amounts of N translocated below the root zone are significantly reduced; (ii) during the major oil-accumulation stage, trees are not exposed to high N levels, which reduces the danger of impaired oil quality [46].

## 14. Fertilization Management Criteria

It is evident that olives have to be fertilized to maintain proper tree nutritional status and not impair growth or production [3]. However, over-fertilization has a potential negative effect on olive yield [14], oil quality [35,46], the environment [9], and, obviously, orchard profitability.

Fertilization regime can be based on one of the three following options. The first is constant application of nutrients according to a preset scheme, without any reference to soil or leaf analyses. A second alternative for the decision-taking process of fertilization is soil analysis. In the case of N, especially in drip-fertigated orchards, soil analysis is not very useful due to inherent soil and system variability and N dynamics in the soil, including its transformation from one form to another, its high mobility, and its interaction with the microbiological population. In the case of P and K, soil analysis is a more useful tool, enabling the grower to obtain information about the nutritional potential of the soil and adjust fertilization accordingly. However, little information is available relating soil P and K levels to tree nutritional status for olive orchards.

The third and most commonly used tool by growers to diagnose the nutritional status of the orchard is leaf analysis [121]. Although accepted and published threshold values are based on rain-fed orchards, they can still be used as a basic reference, to be modified according to new research, for intensive, irrigated orchards. Zipori et al. [13] showed that leaf N concentration is not affected by irrigation level, which indicates that the range of optimal leaf N concentrations, 1.4–1.8%, is valid for both rain-fed and irrigated orchards. There are slight differences between cultivars, but they lie within this range [13,122]. Haberman et al. [14] found maximum yields when leaf N concentrations were within the range of 1.4–1.8%, obtained under annual application rates of 75–150 kg N ha$^{-1}$ over 6 years. At an annual application rate of 300 kg N ha$^{-1}$, only a slight increase in leaf N concentration was observed, which was not proportional to the increase in application level. However, there was a yield reduction at that application rate. It seems that the adequate N application rate for intensive, irrigated olive orchards is between 75 and 150 kg ha$^{-1}$ year$^{-1}$ and that the decision within this range should be taken according to the nutritional status of the trees, based on leaf analysis. Interestingly, vegetative development increases with increasing N application rates, without leveling off, even at the highest annual application rate of 300 kg N ha$^{-1}$. The practical implication of this is that in the first years of the orchard, relatively high N-application rates can be maintained to accelerate vegetative development, which is the basis for future yields. In subsequent years, with bearing trees, application rates can be reduced to avoid yield impairment and reduction in oil quality.

It must be noted that it is almost impossible to identify over-fertilization, especially of N, from leaf analysis data. The relation between application level and leaf mineral concentration bears the pattern of a saturation curve, showing that the higher the application level, the lower the slope of the response line [3]. Hence, there is an urgent need to develop alternative diagnostic tools for excess N. One possibility would be to use fruit pulp analysis, since N in the fruit appears to better represent the tree's N status at excess levels [3]. Threshold values based on this criterion have yet to be developed and evaluated.

The traditionally used sufficiency threshold value for P is 0.1% of leaf dry matter. As already noted, this value is based on observations from rain-fed orchards. In recent studies on irrigated, fertigated trees [41,42], a linear positive yield response to P fertilization was found up to a leaf P concentration of 0.19%, mostly through its influence on reproductive processes. These results emphasize the need to adapt threshold values to management practices. For example, in Israel, growers have adopted the value of 0.13% as the threshold deficiency level for irrigated olives. This value was selected rather intuitively, and needs to be verified and updated by more information from field studies.

As already mentioned, K is the nutrient taken up by olives in the largest amounts. Freeman et al. [50] indicated a value of 0.8% in leaves as the sufficiency level and values below 0.4% as deficiency. As with the other nutrients, the sufficiency level for intensive, irrigated orchards should be updated to leaf K concentrations higher than the present value of 0.8%.

The effect of fruit load on leaf analysis data was studied by Bustan et al. [62], who found that fruit load had no significant effect on leaf N or P concentrations but significantly affected leaf K concentrations, being low in ON years and vice versa. The implication of this finding is that toward an ON year, the nutritional status of the trees should be well taken care of regarding K.

In view of the large differences in tree behavior regarding nutritional status between intensive and non-intensive olive orchards, we suggest an initial modified set of leaf analysis threshold values for intensive orchards: 1.4% for N, 0.13% for P, and 0.9% for K. These values should be verified by additional information from field experiments and adapted to local conditions and cultivars.

## 15. Indirect Environmental Pollution

Environmental damage (to soil, air, and ground and surface water bodies, among others) from over-fertilization of agricultural crops in general, including olive orchards, can be referred to as direct pollution. However, over-fertilization also results in indirect pollution. The fertilizer industry and other supporting systems (transport, application etc.) emit $CO_2$ and other polluting products into the environment. If the produced fertilizers are not absorbed by the plants, the environmental damage is

even magnified. Raw materials for the production of some fertilizers, such as P and K fertilizers, are exhaustible resources. Adaptation of precise fertilization approaches, and recycling of wastes and plant material, will reduce the exhaustion rate of the raw materials.

## 16. Conclusions

Sustainable nutrition of olive orchards has to cover a wide range of considerations. Adequate plant nutrition is a prerequisite for vegetative development, fruit production, and oil quality. At the same time, the extensive use of fertilizers, especially in intensive orchards, increases environmental pollution risks and might impair orchard productivity and oil quality. Careful fertilization management will improve economic and environmental results in olive orchards. Leaf analysis is the most important tool in defining the orchard's nutritional needs. However, for fully irrigated orchards, the threshold values of sufficiency and deficiency have to be thoroughly revised and adapted. Our view is that a completely new set of threshold values should be established for irrigated orchards. An important aspect of sustainability with respect to olive orchard nutrition is recycling. Recycling reduces the potential negative impact of wastes, reduces costs, and has additional benefits such as soil conservation, soil organic matter improvement, soil water retention, and more. A fertilization strategy based on environmental and economic considerations should address the recycling of pruned material and olive industry waste, as well as careful utilization of wastewater. Combining this approach with cover crops and systematic leaf and soil analyses should enhance the efficient utilization of fertilizers while maintaining high yields and oil quality.

**Author Contributions:** I.Z.—Soil-plant-water relationships and orchard management. R.E. and U.Y.—Plant nutrition and chemical soil properties. A.B.-G.—Olive irrigation and soil physics. A.D.—Olive physiology and biology and orchard management. All authors have read and agreed to the published version of the manuscript.

**Funding:** This research received no external funding.

**Conflicts of Interest:** The authors declare no conflict of interest.

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
