# Peer review of "Sustainable Management of Olive Orchard Nutrition: A Review"

_agriculture, doi:10.3390/agriculture10010011_

Round 1

Reviewer 1 Report

The manuscript corresponds well to the profile of "Agriculture". It has a well-suited title in relation to its content.

The abstract fulfills its task and reflects well the content of the manuscript. The aim of the study is not clearly defined, thus, it is not possible accurately summarize these researches. The manuscript layout contains the appropriate subtitles as it is required for the review papers. It is transparent and good-composed according to the requirements of "Agriculture". The research results were presented reasonably, both in the text, tables, figures and photographs. The multi-aspect Discussion is interesting and supported by a reasonable and extensive selection of literatures. The summary is logical but it does not reflect the aim of the study (which is not clearly defined in this manuscript). The literature is well chosen.

This manuscript presents a proper approach to nutrient management showing both the deficiencies and their excesses, and also environmental effects of using various fertilizer variants (mineral and natural fertilizers and irrigation) in an olive orchard. The authors of this manuscript indicate the need for full fertilizer diagnostics based on the creation of new diagnostics taking into account the all elements of orchard fertilization (the analyses of leaf and soil in relation to the applied fertilization, irrigation, covering, and the use of organic fertilizers).

This manuscript contains the current and practical aspects. The few errors have been highlighted in the text and they should be corrected.

Reviewer 2 Report

The paper is complete and well organised. A few adjustments in my view should be done.

Table 1 indicates data from 2 different sets of years, but discussion treats them as homogeneous. Either the description should be more distinctive, or the figures and years be adjusted.

Confusion is made, for the same Table 1, between the three year and one year period. Thus fertilisation is described as annual (line 61); total annual amount removed is instead the sum of the 3 years, as from the table. The problem shows up also in lines 357-358.

Figure 3: not all treatments acronyms carry the A or S indication. It is a very minor issue, but I think the 2 figures (2 and 3) should be homogeneous in the format.

Lines 242-244: the amount of compost applied is indicated per tree, but the contribution of the compost as concerns the individual elements is indicated per ha. Therefore the total amount of compost per Ha cannot be understood. The amount per Ha of compost should be indicated, or, subordinately, the number of trees per Ha.

Alignment of references should be uniformed
